# Multimerization strategies for efficient production and purification of highly active synthetic cytokine receptor ligands

**Sofie Mossner[1], Hoang T. Phan[2], Saskia Triller[2], Jens M. Moll[1], Udo Conrad[2], Jürgen Scheller**[1] *

**1** Institute of Biochemistry and Molecular Biology II, Medical Faculty, Heinrich-Heine-University, Düsseldorf, Germany, **2** Leibniz Institute of Plant Genetics and Crop Plant Research (IPK), Stadt Seeland, Gatersleben, Germany

* jscheller@uni-duesseldorf.de

**Data Availability Statement:** All relevant data are within the paper and its Supporting Information files.

## Abstract

Cytokine signaling is transmitted by cell surface receptors which act as natural biological switches to control cellular functions such as immune reactions. Recently, we have designed synthetic cytokine receptors (SyCyRs) consisting of green fluorescent protein (GFP)- and mCherry-nanobodies fused to the transmembrane and intracellular domains of cytokine receptors. Following stimulation with homo- and heterodimeric GFP-mCherry fusion proteins, the resulting receptors phenocopied signaling induced by physiologically occurring cytokines. GFP and mCherry fusion proteins were produced in *E. coli* or CHO-K1 cells, but the overall yield and stability was low. Therefore, we applied two alternative multimerization strategies and achieved immunoglobulin Fc-mediated dimeric and coiled-coil GCN4pII-mediated trimeric assemblies. GFP- and/or mCherry-Fc homodimers activated synthetic gp130 cytokine receptors, which naturally respond to Interleukin 6 family cytokines. Activation of these synthetic gp130 receptors resulted in STAT3 and ERK phosphorylation and subsequent proliferation of Ba/F3-gp130 cells. Half-maximal effective concentrations ($EC_{50}$) of 8.1 ng/ml and 0.64 ng/ml were determined for dimeric GFP-Fc and mCherry-Fc, respectively. This is well within the expected $EC_{50}$ range of the native cytokines. Moreover, we generated tetrameric and hexameric GFP-mCherry-Fc fusion proteins, which were also biologically active. This highlighted the importance of close juxtaposition of two cytokine receptors for efficient receptor activation. Finally, we used a trimeric GCN4pII motif to generate homo-trimeric GFP and mCherry complexes. These synthetic cytokines showed improved $EC_{50}$ values ($GFP^3$: 0.58 ng/ml; $mCherrry^3$: 0.37 ng/ml), over dimeric Fc fused variants. In conclusion, we successfully generated highly effective and stable multimeric synthetic cytokine receptor ligands for activation of synthetic cytokine receptors.

**Funding:** This work was funded by a grant from the Deutsche Forschungsgemeinschaft (SFB1116).

**Competing interests:** The authors have declared that no competing interests exist.

**Abbreviations:** (except for standard chemicals and techniques) IL-, interleukin; gp130, glycoprotein 130 kDa; IL-6R, interleukin 6-receptor; STAT, signal transducer and activator of transcription; GFP, green fluorescent protein.

# Introduction

Cytokines control immune responses but are also involved in homeostatic processes such as development, differentiation, growth and regeneration. Signal transduction of cytokines is executed by natural biological switches which among many other functions control immune related processes [1]. Cytokines switch transmembrane receptors from the off-state into the on-state via receptor dimerization or multimerization. The on-state might be interrupted by negative feedback mechanisms or depletion of the cytokine and cytokine receptor. Recently, we have designed synthetic cytokine receptors (SyCyRs) which phenocopy IL-6 and IL-23 signaling via homodimeric gp130 and heterodimeric IL-23R/IL-12Rbeta1 receptors [2]. SyCyRs incorporate nanobodies specifically recognizing GFP or mCherry [3, 4] fused to transmembrane and intracellular receptor domains. The nanobodies serve as extracellular sensors for homo- and heteromeric GFP-mCherry fusion proteins which induce receptor dimerization. A nanobody or VHH domain consists of the N-terminal variable domain of a Camelidae heavy chain antibody which is sufficient for antigen binding [5]. Synthetic cytokine receptors might become important tools for immunotherapeutic applications [6] with Chimeric Antigen Receptor (CAR) T-cell therapy being the first example which has been approved as gene therapy for the treatment of severe cases of acute lymphatic leukemia [7].

Moreover, synthetic cytokine biology can decipher the potential of cytokine receptor crosstalk. In a reductionistic view, a cytokine binds only to its corresponding cytokine receptor complex which is composed either of receptor homo- or heterodimers. This simple view has been challenged for many cytokines and cytokine receptors which have multiple binding partners. For example, the dimerization of two gp130 receptor chains is essential for IL-6 and IL-11 signal transduction. Furthermore, gp130 acts as a co-receptor for IL-27, CNTF, CT-1, LIF and OSM. On the other hand, IL-35 from the IL-12-type cytokine family was proposed to activate a variety of different receptor complexes, including gp130 homodimers, IL-12Rbeta2 homodimers and gp130/IL-12Rbeta2 heterodimers [8]. Using chimeric cytokine receptors, we have shown that gp130 can form biologically active complexes with IL-23R and IL-12Rbeta2 of the closely related IL-12-type cytokine family [9].

For the analysis of synthetic cytokine receptor signaling, large quantities of stable and biologically active synthetic cytokine receptor ligands are required. Cytokines have two or more binding sites for the corresponding receptors and mainly form complexes consisting of two homo- or heterodimerized receptors [10]. Therefore, we applied two different strategies to generate dimeric and multimeric synthetic cytokine receptor ligands. GFP and mCherry were expressed in frame with the Fc portion of an IgG antibody to generate dimeric ligands. Fusion proteins of GFP and mCherry with trimeric GCN4pII motif were utilized to produce trimeric ligands. The resulting fusion proteins were expressed, purified and functionally characterized using the established SyCyR(IL-6) as read out system. The Fc-part from IgG antibodies is widely used in biotechnology. Either as an efficient purification tag, which facilitates one-step purification of Fc-fusion proteins, as antibodies via Protein A sepharose or as a dimerization tool [11]. Here, we used the Fc-tag in two ways, to simplify purification and as a dimerization tool. Many proteins oligomerize *via* short coiled-coil sequences containing seven-amino acid repeats (heptad repeat). A well-studied α-helical coiled-coil protein sequence is the dimerization domain of the yeast transcription factor GCN4 leucine zipper. When Val and Leu residues at a and d heptad positions, respectively, were substituted by isoleucines, the 33-residue GCN4 leucine zipper dimer was switched to a trimeric assembly, termed the GCN4-pII [12]. Recently, trimeric GCN4-pII has been used to trimerize the influenza ectodomain hemagglutinin [13, 14].

## Materials and methods

### Cells and reagents

All cells were grown at 37˚C with 5% $CO_2$ in a water saturated atmosphere in Dulbecco's modified Eagle's Medium (DMEM) high glucose culture medium (GIBCO®, Life Technologies, Darmstadt, Germany) with 10% fetal calf serum (GIBCO®, Life Technologies, Darmstadt, Germany) and 60 mg/l penicillin and 100 mg/l streptomycin (Genaxxon Bioscience GmbH, Ulm, Germany). Murine Ba/F3-gp130 cells were obtained from Immunex (Seattle, WA, USA) and grown in the presence of Hyper-IL-6, a fusion protein of IL-6 and soluble IL-6 receptor [15]. 0.2% (10 ng/ml) of conditioned medium from a stable clone of CHO-K1 cells secreting Hyper-IL-6 in the supernatant (stock solution approximately 5 μg/ml as determined by ELISA) were used to maintain Ba/F3-gp130 cells and derivates thereof. CHO-K1 (ACC-110) cells were provided from Leibniz Institute DSMZ-German Collection of Microorganisms and Cell Cultures (Braunschweig, Germany). Phoenix-Eco cell line was received from Ursula Klingmüller (DKFZ, Heidelberg, Germany). Phospho-STAT3 (Tyr705; D3A7; cat. #9145; 1:1000), STAT3 (124H6; cat. #9139; 1:1000), phospho-p44/42 MAPK (ERK1/2; Thr-202/Tyr-204; D13.14.4E; cat. #4370; 1:1000), p44/42 MAPK (ERK1/2; cat. #9102; 1:1000), GFP (4B10; cat. #2955, 1:1000), Myc-Tag (71D10; cat. #2278; 1:1000) and HA-Tag (C29F4; cat. #S724S; 1:1000) monoclonal antibodies (mAbs) were sustained from Cell Signaling Technology (Frankfurt, Germany). mCherry (PA5-34974, 1:1000) polyclonal antibody was sustained from Invitrogen (Carlsbad, CA, United States). Human IgG1 Fc (cat. #31423, 1:5000) and peroxidase-conjugated secondary mAbs (cat. #31432 and #31462; 1:2500) were obtained from Pierce (Thermo Fisher Scientific, St. Leon-Rot, Germany). Alexa Flour 488 conjugated Fab goat anti-rabbit IgG (cat. #A11070; 1:500) was received from Thermo Fisher Scientific (Waltham, MA, USA).

### Composition of SyCyRs and synthetic fluorescent ligands

SyCyR cDNAs subcloned in pcDNA3.1 incorporated the IL-11R signal peptide (Q14626, aa 1–22), a myc tag (EQKLISEEDL; SyCyRs containing GFP$_{VHH}$) or a FLAG tag (DYKDDDDK) and HA tag (YPYDVPDYA; for SyCyRs containing mCherry$_{VHH}$) followed by a nanobody (GFP$_{VHH}$ or mCherry$_{VHH}$), some residues of the extracellular domain (ECD), the complete transmembrane (TMD) and complete intracellular domain (ICD) of gp130. The included amino acids for the human gp130 receptor were aa T607-Q918 (Q17RA0; comprising 14 aa of the ECD) (sequence shown in S1A Fig). The pcDNA3.1 plasmids were digested with PmeI (MssI) and the SyCyR cDNAs were ligated into pMOWS (retroviral plasmid). Ba/F3-gp130 cells were consecutively retrovirally transduced with two SyCyRs using two different pMOWS plasmids with either puromycin (for GFP$_{VHH}$-SyCyRs) or hygromycin resistance (for mCherry$_{VHH}$-SyCyRs) for selection of stably transduced clones. Synthetic ligands were expressed in CHO-K1 cells as previously described [2].

Synthetic Fc-tagged ligands (sequence shown in S1B Fig) with neomycin resistance gene were stably expressed in CHO-K1 cells using single clone selection with G418 (Genaxxon, Ulm, Germany). After transfection cells were cultivated with G418 for two weeks and single clone selection was carried out. 0.5 cells/well were seeded out and single clone colonies screened for protein expression. The colony expressing and secreting the most protein was then cultured for the ligand production in roller bottles.

### Transfection and selection of cells

Ba/F3-gp130 cells were retrovirally transduced using pMOWS plasmids coding for different SyCyRs as described previously [2]. The packaging cell line was Phoenix-Eco. After

transduction cells were grown as described above and supplemented with puromycin (1.5 μg/ml) and/or hygromycin B (1 mg/ml) (Carl Roth, Karlsruhe, Germany). CHO-K1 cells were stably transfected with TurboFect[TM] (Thermo Fisher Scientific, Waltham, United States) and then selected using 1.125 mg/ml G-418 sulfate (Genaxxon, Ulm, Germany).

## Expression and purification of synthetic ligands

GFP-Fc, mCherry-Fc, GFP-mCherry-Fc and 2xGFP-mCherry-Fc were purified from CHO-K1 cell culture supernatant obtained from stably transfected cells. Cells were cultured in a rollerbottle system (IBS integra bioscience, Zizers, Switzerland) with 10% low IgG fetal calf serum (GIBCO®, Life Technologies, Darmstadt, Germany) DMEM for two months. The supernatant was collected every 3–4 days. 1 L of supernatant was collected and Fc-tagged proteins purified using MabSelect[TM] HiTrap[TM] columns (GE Healthcare, Chalfont St Giles, UK). Proteins were eluted by pH shift using citrate buffer of pH 5.5 and 3.2. Buffer exchange to PBS was achieved using NAP[TM]-25 columns (GE Healthcare, Chalfont St Giles, UK).

## Agro-infiltration

Agro-infiltration for the expression of recombinant proteins was described in detail by Phan and Conrad [16] and is briefly described here. Agrobacteria harboring shuttle vectors to express recombinant GFP[3] and mCherry[3] (Fig 5A) and Agrobacteria harboring a plant vector to express HcPro, a suppressor of gene silencing, were pre-cultivated separately in LB medium with 50 μg/ml kanamycin, 50 μg/ml carbenicillin and 50 μg/ml rifampicin overnight at 28˚C and 140 rpm. The pre-cultures were added to new LB medium containing the appropriate antibiotics. After cultivation for 24 h, bacteria were harvested by centrifugation (4 000 g, 30 min, 4˚C) and resuspended in infiltration buffer (10 mM of 2-(N-morpholino) ethanesulfonic acid (MES), 10 mM of $MgSO_4$, pH 5.6). Agrobacteria harboring the shuttle vector for recombinant protein expression and the plant vector to express HcPro were combined and diluted in infiltration buffer to a final $OD_{600}$ of 1.0. *N. benthamiana* plants (six to eight weeks old) were infiltrated by completely submerging each plant in an agrobacterium-containing cup standing inside a desiccator. A vacuum was applied for 2 min and then quickly released. The plants were then grown in the greenhouse at 21˚C under 16 h of light per day. Five days after infiltration, leaf samples were harvested and stored at −80˚C.

## GFP[3] and mCherry[3] purification by IMAC from plants

Five days after vacuum infiltration, leaf samples were harvested, frozen in liquid nitrogen and homogenized using a commercial blender. Proteins from 80 g of infiltrated leaves were extracted in 240 ml of 50 mM Tris buffer (pH 8.0). The extracts were clarified by centrifugation (75 600 g, 30 min, 4˚C) and then filtered through paper filters. The clarified extracts were mixed with 20 ml of packed Ni-NTA agarose resin that had previously been washed twice with water. After mixing for 30 min at 4˚C, the mixture was added to a chromatography column. Thereafter, the column was extensively washed (50 mM of $NaH_2PO_4$, 300 mM of NaCl, 30 mM of imidazole, pH 8.0). Recombinant proteins were eluted from the column with elution buffer (50 mM of $NaH_2PO_4$, 300 mM of NaCl, 125 mM of imidazole, pH 8.0), placed in dialysis bags, concentrated in PEG 6 000 and dialyzed against PBS.

## Cross-linking reaction

A cross-linking reaction was performed to determine the multimeric state of the plant-derived GFP[3] and mCherry[3] proteins following the method described by Phan and co-workers [13].

Briefly, one µg of purified plant-derived proteins was mixed with Bis[sulfosuccinimidyl] suberate (BS3) to a 5 mM final concentration and incubated for 30 min at room temperature. The cross-linking reaction was stopped by the addition of 1 M Tris-HCl pH 8.0 to a final concentration of 50 mM and incubated for 15 min at room temperature. After cross-linking, proteins were separated on a 10% SDS-PAGE under reducing conditions, blotted and analyzed by Western blot using anti-c-myc monoclonal antibody.

## Cell viability assay

Ba/F3-gp130 cell lines were washed 3 times in sterile PBS to remove cytokines from the medium. $5x10^4$ cells were suspended in DMEM containing 10% FCS, 60 mg/l penicillin and 100 mg/ml streptomycin. Cells were cultured for 3 d in a volume of 100 µl with or without cytokine/synthetic ligands. The CellTiter Blue Viability Assay (Promega, Karlsruhe, Germany) was used to determine the approximate number of viable cells by measuring the fluorescence (excitation 560 nm, emission 590 nm) using the Infinite M200 Pro plate reader (Tecan, Crailsheim, Germany). After adding 20 µl per well of CellTiter Blue reagent (point 0) and up to 2 h fluorescence was measured approximately every 20 min. For each condition of an experiment 3–4 wells were measured. All values were normalized by subtracting time point 0 values from the final measurement.

## Stimulation assays

Cells were washed 4 times with sterile PBS to remove cytokines. After starvation for 4 h in serum-free DMEM, the cells were stimulated with 100 ng/ml of purified protein as indicated for 15 min. Cells were harvested and frozen in liquid nitrogen. For analysis, the cells were lysed in lysis buffer containing 10 mM Tris-HCl, pH 7.8, 150 mM NaCl, 0.5 mM EDTA, 0.5% NP-40, 1 mM sodium vanadate, 10 mM $MgCl_2$ and a complete, EDTA-free protease inhibitor cocktail tablet (Roche Diagnostics, Mannheim, Germany) for 2 h. Protein concentration was measured by BCA Protein assay (Pierce, Thermo Fisher Scientific, Waltham, USA) following the instructions of the manufacturer. Analysis of protein activation and expression as indicated above was done by immunoblotting using 50 µg of each lysate.

## Western blotting

Defined amounts of proteins were loaded per lane, separated by SDS-PAGE under reducing and non-reducing conditions and transferred to a polyvinylidene fluoride (PVDF) membrane (Carl Roth, Karlsruhe, Germany). The membranes were blocked in 5% fat-free dried skimmed milk (Carl Roth, Karlsruhe, Germany) in TBS-T (10 mM Tris-HCl pH 7.6, 150 mM NaCl, 0,5% Tween-20) for 4 h. After blocking, primary antibodies diluted in 5% fat-free dried skimmed milk in TBS-T (STAT3, ERK, Fc) or 5% bovine serum albumin in TBS-T (pSTAT3, pERK, HA, myc, GFP, mCherry) were added and incubated at 4˚C overnight. After washing with TBS-T, the membranes were either incubated with secondary peroxidase-conjugated antibodies in 5% fat-free dried skimmed milk in TBS-T at room temperature for at least 1 h or directly detected (Fc antibody). Signal detection was achieved using the ECL Prime Western Blotting Detection Reagent (GE Healthcare, Freiburg, Germany) and the Chemo Cam Imager (INTAS Science Imaging Instruments, Göttingen, Germany). For a second round of detection, the membranes were stripped in 62.5% Tris-HCl, pH 6.8, 2% SDS, 0.1% β-mercaptoethanol at 60˚C for 30 min and then blocked again in 5% fat-free dried skimmed milk in TBS-T for at least 3 h before using the next primary antibody.

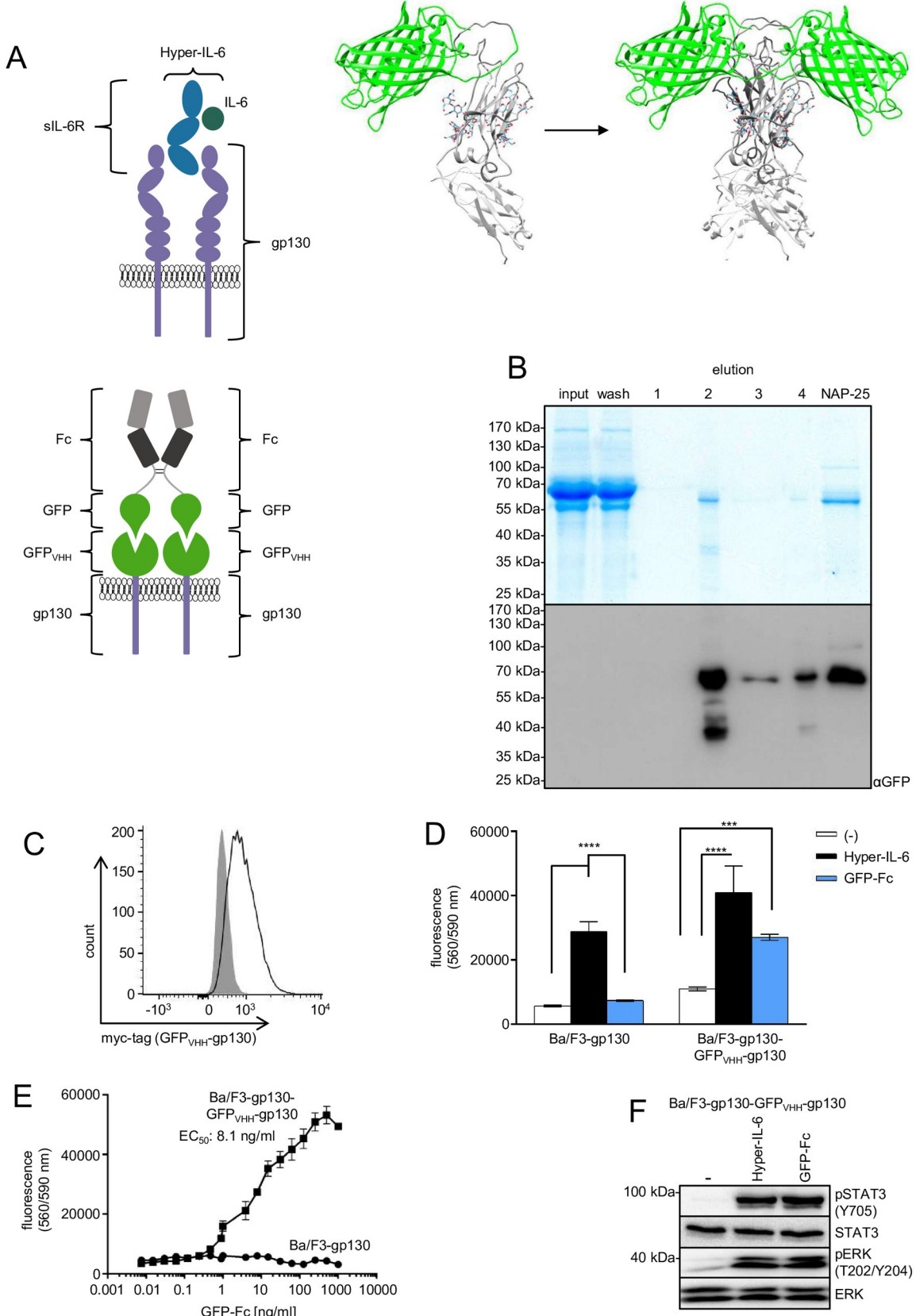

**Fig 1. Purification of synthetic GFP-Fc cytokine and simulation of IL-6 trans-signaling via GFP$_{VHH}$-gp130-SyCyRs.** (A) Schematic illustration of IL-6 trans-signaling and a phenocopy using a synthetic receptor ligand system. The complex of IL-6 (dark green) and

soluble IL-6R (blue) binds to two gp130 receptors (purple) and activates an intracellular signaling cascade. In the synthetic system, a GFP-Fc fusion protein (green and grey) forms a dimer via the Fc domain of an IgG antibody and binds to GFP$_{VHH}$-gp130-SyCyR (green and purple). The GFP$_{VHH}$ is fused to 13 aa of the ECD, TMD and ICD of gp130. GFP (green)-Fc (grey) was modeled using Phyre2. The resulting coordinates were superpositioned onto a structure of Fc fragment (PDB 4CDH) using UCSF Chimera. The image is similar to the original image and for illustrative purposes only. (B) Coomassie gel and Western blot with anti-GFP antibody. CHO-K1 cells were stably transduced with the cDNA coding for GFP-Fc fusion protein. The supernatant with the secreted protein of 58 kDa was collected and purified via Protein A chromatography. (C) Cell surface expression of GFP$_{VHH}$-gp130-SyCyR (solid line) in Ba/F3-gp130 cells detected by myc-antibody. Grey-shade areas indicate not transfected Ba/F3-gp130 cells (negative control). (D) Proliferation of Ba/F3-gp130 and Ba/F3-gp130-(GFP$_{VHH}$-gp130) cells without cytokine (-), in presence of 10 ng/ml Hyper-IL-6 or in presence of 100 ng/ml GFP-Fc. One representative experiment out of four is shown. (E) Proliferation of Ba/F3-gp130-(GFP$_{VHH}$-gp130) and Ba/F3-gp130 cells with increasing concentrations of 0.0004–1000 ng/ml GFP-Fc. One representative experiment out of four is shown. (F) STAT3 and ERK1/2 activation in Ba/F3-gp130-(GFP$_{VHH}$-gp130) cells treated either with 10 ng/ml Hyper-IL-6 or 100 ng/ml GFP-Fc for 15 min. Equal amounts of protein (50 μg per lane) were analyzed via specific antibodies detecting phospho-STAT3/ERK1/2 and STAT3/ERK1/2. Western blot data shows one representative experiment out of four.

## Coomassie staining

5 μg of protein were loaded per lane and separated by SDS-PAGE under reducing and non-reducing conditions. The gel was stained with Coomassie staining solution (80% ethanol, 20% acetic acid, 4% Coomassie brilliant blue R250) for 20 min and washed twice with $H_2O$. The gel was then destained overnight in destaining solution (20% ethanol, 10% acetic acid) and scanned for analysis.

## Cell surface detection of receptors

For the detection of the SyCyR cell surface expression of stably transfected Ba/F3-gp130 cells, cells were washed in FACS buffer (PBS, 1% bovine serum albumin). $5 \times 10^5$ cells were resuspended in 50 μl FACS buffer containing specific primary antibodies as indicated (myc 1:100; HA 1:1000). Cells were incubated for 1 h at room temperature and then washed with FACS buffer and resuspended in 50 μl FACS buffer with Alexa Flour 488 conjugated Fab goat anti-rabbit IgG (cat. #A11070) in a 1:500 dilution and again incubated for 1 h at room temperature. Cells were washed with FACS buffer, resuspended in 500 μl FACS buffer and analyzed by flow cytometry (BD FACSCanto II flow cytometer, BD Biosciences, San Jose, CA, USA). Data was evaluated using the FCS Express 4 Flow software (De Novo Software, Los Angeles, CA, USA).

## Modeling

Protein models were generated via Phyre2 (PMID 25950237). Molecular graphics and structural super positioning of models with PDB: 4CDH, 4EUL, 5KKV were generated using UCSF Chimera version 1.13.1, developed by the Resource for Biocomputing, Visualization, and Informatics at the University of California, San Francisco, with support from NIH P41-GM103311 (PMID: 15264254).

## Statistical analysis

Data are shown as mean ±SD. Multiple comparisons were determined with GraphPad Prism 6 (GraphPad Software, San Diego, CA, USA) using one-way ANOVA column analyses. Statistical significance was set to $p < 0.05$ (**** $p < 0.0001$, *** $p < 0.001$, ** $p < 0.01$, * $p > 0.1$).

## Results and discussion

### Dimeric GFP-Fc and mCherry-Fc fusion proteins are effective activators of synthetic cytokine receptors

gp130 is naturally activated by IL-6 either in complex with the membrane bound IL-6R or the soluble IL-6R (Fig 1A). In the synthetic gp130 cytokine receptor (GFP$_{VHH}$-gp130), the

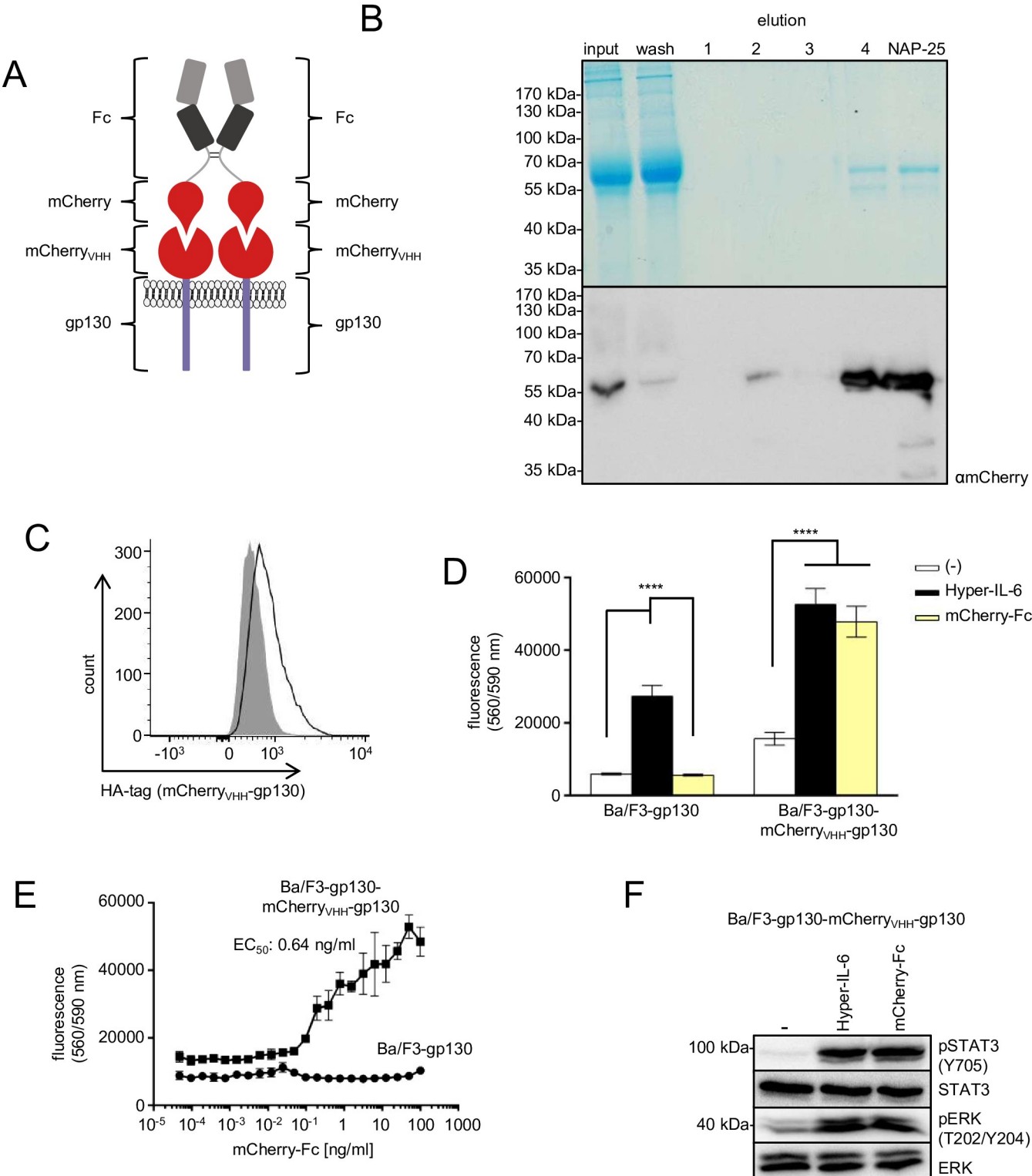

**Fig 2. Purification of synthetic mCherry-Fc cytokine and simulation of IL-6 trans-signaling via mCherry$_{VHH}$-gp130-SyCyR.** (A) Schematic illustration of dimeric synthetic cytokine mCherry-Fc binding to two mCherry$_{VHH}$-gp130-SyCyRs and thereby activating the signaling cascade. The image is similar to the original image and for illustrative purposes only. (B) Coomassie gel and Western blot with anti-mCherry antibody. CHO-K1 cells were stably transduced with the cDNA coding for mCherry-Fc fusion protein. The supernatant with the secreted protein of 57 kDa was collected and purified via Protein A affinity chromatography. (C) Cell surface expression of mCherry$_{VHH}$-gp130 SyCyR (solid line) in Ba/F3-gp130 cells detected by HA-antibody. Grey-shade areas indicate not transfected Ba/F3-gp130 cells (negative control). (D) Proliferation of Ba/F3-gp130 and Ba/F3-gp130-(mCherry$_{VHH}$-gp130) cells without cytokine

(-), in presence of 10 ng/ml Hyper-IL-6 or in presence of 100 ng/ml mCherry-Fc. One representative experiment out of four is shown. (E) Proliferation of Ba/F3-gp130-(mCherry$_{VHH}$-gp130) and Ba/F3-gp130 cells with increasing concentrations of 0.0004–1000 ng/ml mCherry-Fc. One representative experiment out of four is shown. (F) STAT3 and ERK1/2 activation in Ba/F3-gp130-(mCherry$_{VHH}$-gp130) cells treated either with 10 ng/ml Hyper-IL-6 or 100 ng/ml mCherry-Fc for 15 min. Equal amounts of protein (50 μg per lane) were analyzed via specific antibodies detecting phospho-STAT3/ERK1/2 and STAT3/ERK1/2. Western blot data shows one representative experiment out of four.

extracellular domain of gp130 was replaced by a nanobody specifically binding to GFP (Fig 1A, S1A Fig). To generate a corresponding synthetic ligand, we fused GFP to the Fc-part of an IgG-antibody connected by a flexible peptide linker containing a TEV-cleavage site (Fig 1A, S1B Fig). CHO-K1 cells were stably transduced with the cDNA coding for the GFP-Fc fusion protein. High expressing single clones were selected and expanded for protein expression. GFP-Fc fusion proteins were purified from the CHO-K1 cell culture supernatant by Protein A-sepharose chromatography (Fig 1B). The overall yield of GFP-Fc was 1.23 mg/l, which is in the expected range for non-optimized standard laboratory expression systems. The protein was stable at 4˚C and could be frozen and thawed without loss of protein concentration and biological activity. SDS-PAGE analysis followed by Coomassie staining was performed under reducing and non-reducing conditions. Demonstrating a duplication of the molecular weight due to Fc dimerization under non-reducing conditions (S2A Fig). Ba/F3 cells stably expressing GFP$_{VHH}$-gp130 and natural gp130 on the cell surface (Fig 1C) were used to analyze the biological activity of purified GFP-Fc fusion proteins. Murine Ba/F3 cells expressing human gp130 proliferate after supplementation with IL-6 and the sIL-6R or Hyper-IL-6, which is a fusion protein of IL-6 and sIL-6R [15]. Importantly, proliferation of Ba/F3-gp130-(GFP$_{VHH}$-gp130) cells was stimulated by GFP-Fc and Hyper-IL-6, whereas proliferation of Ba/F-gp130 was only induced by Hyper-IL-6 but not by GFP-Fc (Fig 1D). The half-maximal effective concentration of GFP-Fc to induce cellular proliferation was determined to be 8.1 ng/ml (Fig 1E), which is in good agreement with previous findings using dimeric GFP-GFP-fusion proteins from CHO-K1 cell culture supernatants [2]. The EC$_{50}$ of purified Hyper-IL-6-Fc or Hyper-IL-6 cell culture supernatant were 0.7 ng/ml and 0.16 ng/ml, respectively (S2B Fig). Proliferation of Ba/F3-gp130-(GFP$_{VHH}$-gp130) cells is dependent on STAT3 and ERK1/2 phosphorylation, hence we verified that GFP-Fc and Hyper-IL-6 induced comparable phosphorylation of STAT3 and ERK1/2 (Fig 1F).

In addition, we used Ba/F3 cells, expressing mCherry$_{VHH}$-gp130 and gp130 to test the biological activity of dimeric mCherry-Fc (Fig 2A and 2C, S1A and S1B Fig). mCherry-Fc was produced in stably transfected CHO-K1 cells. Following purification of mCherry-Fc an overall yield of 1.23 mg/l of cell culture supernatant was achieved (Fig 2B). Dimerization of mCherry-Fc was shown by SDS-PAGE followed by Coomassie staining and Western blotting under reducing and non-reducing conditions (S2A Fig). mCherry-Fc efficiently induced proliferation of Ba/F3-gp130-(mCherry$_{VHH}$-gp130) cells, whereas proliferation of Ba/F3-gp130 cells was only induced by Hyper-IL-6 but not mCherry-Fc (Fig 2D). The EC$_{50}$ for proliferation of Ba/F3-gp130-(mCherry$_{VHH}$-gp130) cells was 0.64 ng/ml (Fig 2E) and about 12fold lower compared to GFP-Fc but in the same range as Hyper-IL-6. This is likely explained by the higher affinity of mCherry to mCherry$_{VHH}$ (K$_D$ 0.18–63 nM) [3] compared to the binding of GFP to GFP$_{VHH}$ (K$_D$ ~1 nM) [17]. Importantly, STAT3 and ERK1/2 phosphorylation was induced to a similar extend by mCherry-Fc and Hyper-IL-6 (Fig 2F).

Recently, we described *E. coli* produced multimeric GFP and mCherry fusion proteins able to activate SyCyRs. However, fusion proteins were contaminated with shorter fragments and degradation products which makes it unattractive to use *E. coli* as production host for these multimeric fusion proteins (S3A–S3F Fig). Hence, we decided to express Fc fusion proteins of GFP and mCherry in eukaryotic cells. The production of these multimeric GFP and mCherry

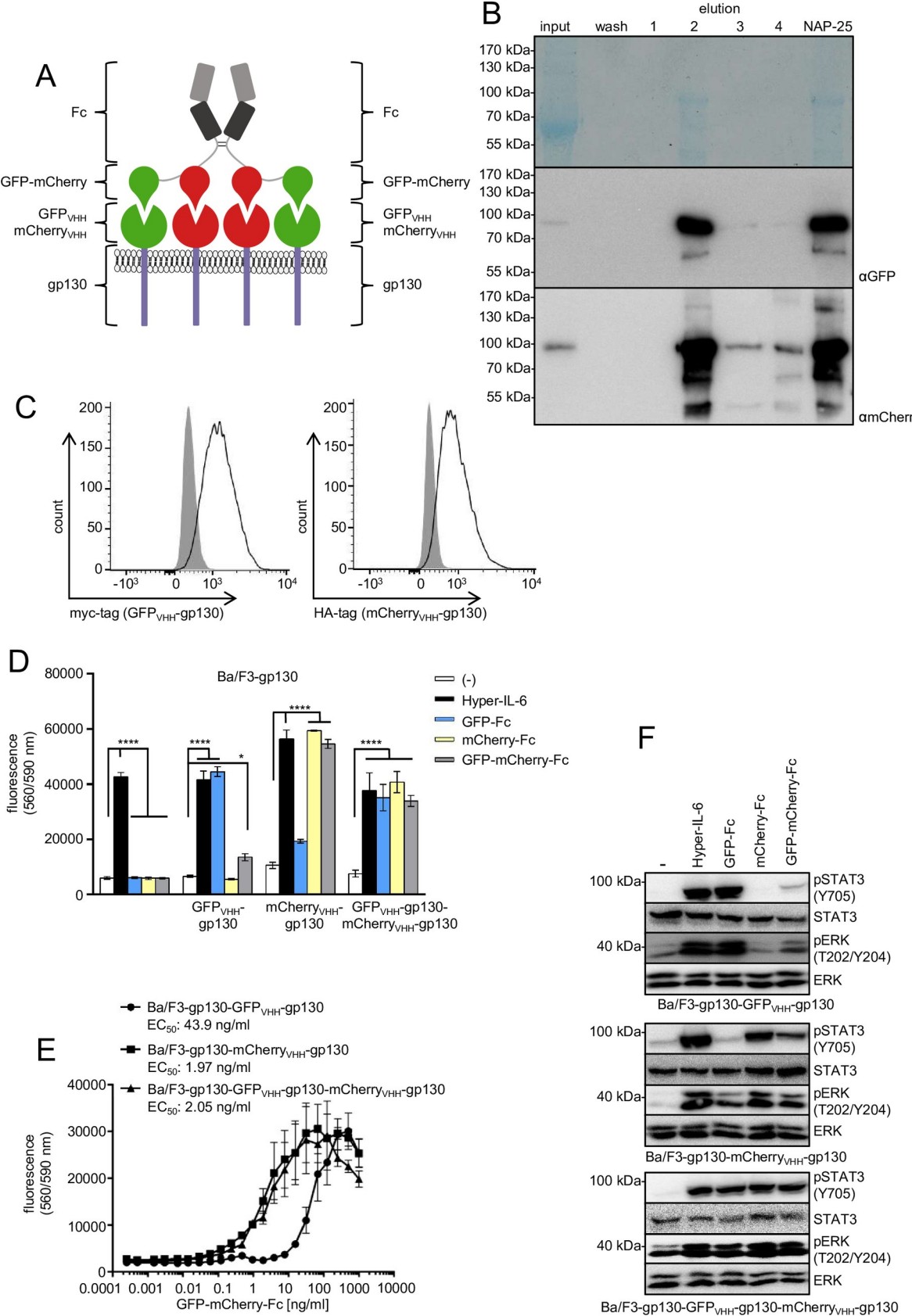

**Fig 3. Purification of synthetic GFP-mCherry-Fc and analysis of signal transduction induced through GFP$_{VHH}$-gp130 and mCherry$_{VHH}$-gp130 SyCyRs.** (A) Schematic illustration of a GFP-mCherry fusion protein linked to the Fc domain of an IgG antibody.

The Fc domain forms a dimer leading to a tetrameric assembly with two central mCherry proteins and GFP subunits on the outside. The synthetic ligand can bind to GFP$_{VHH}$-gp130 and mCherry$_{VHH}$-gp130 SyCyRs or both. The image is similar to the original image and for illustrative purposes only. (B) Coomassie gel and Western blot with anti-GFP and anti-mCherry antibodies. CHO-K1 cells were stably transduced with the cDNA coding for the GFP-mCherry-Fc fusion protein. The supernatant containing the secreted protein with an expected molecular weight of 84.5 kDa was collected and purified via Protein A affinity chromatography. (C) Cell surface expression of GFP$_{VHH}$-gp130 and mCherry$_{VHH}$-gp130 (solid line) in Ba/F3-gp130 cells was detected by myc-antibody (GFP$_{VHH}$) or HA-antibody (mCherry$_{VHH}$). Grey-shaded areas indicate untransfected Ba/F3-gp130 cells (negative control). (D) Proliferation of Ba/F3-gp130, Ba/F3-gp130-(GFP$_{VHH}$-gp130), Ba/F3-gp130-(mCherry$_{VHH}$-gp130) and Ba/F3-gp130-(GFP$_{VHH}$-gp130)-(mCherry$_{VHH}$-gp130) cells without cytokine (-), in presence of 10 ng/ml Hyper-IL-6 or in presence of 100 ng/ml GFP-Fc, mCherry-Fc or GFP-mCherry-Fc. One representative experiment out of four is shown. (E) Proliferation of Ba/F3-gp130-(GFP$_{VHH}$-gp130), Ba/F3-gp130-(mCherry$_{VHH}$-gp130) and Ba/F3-gp130-(GFP$_{VHH}$-gp130)-(mCherry$_{VHH}$-gp130) cells with increasing concentrations of 0.0004–1000 ng/ml GFP-mCherry-Fc. One representative experiment out of three is shown. (F) STAT3 and ERK1/2 activation of Ba/F3-gp130-(GFP$_{VHH}$-gp130), Ba/F3-gp130-(mCherry$_{VHH}$-gp130) and Ba/F3-gp130-(GFP$_{VHH}$-gp130)-(mCherry$_{VHH}$-gp130) cells treated either with 10 ng/ml Hyper-IL-6 or 100 ng/ml GFP-Fc, mCherry-Fc or GFP-mCherry-Fc for 15 min. Equal amounts of protein (50 μg per lane) were analyzed via specific antibodies detecting phospho-STAT3/ERK1/2 and STAT3/ERK1/2. Western blot data shows one representative experiment out of four.

proteins in CHO-K1 cells did, with the exception of mCherry fusion proteins, not result in generation of major shorter fragments and degradation products, even after storage at 4°C for five days (S4A and S4B Fig), suggesting that it is possible to produce these fusion proteins with high stability. Therefore, production in CHO-K1 cells was highly recommended and the addition of the Fc-tag was advantageous for purification purposes. Moreover, in the case of single GFP-Fc and mCherry-Fc fusion proteins, we also demonstrated that the dimerization of GFP and mCherry via the Fc part of an IgG antibody results in spatial distances of both protein subunits that allow efficient dimerization and activation of synthetic cytokine receptors.

In conclusion, we demonstrated that synthetic GFP- and mCherry-Fc fusion proteins were efficient activators of our synthetic cytokine system. Further, they displayed improved stability and allowed for a simplified purification strategy as compared to the originally used GFP-GFP and mCherry-mCherry fusion proteins.

## Tetrameric and hexameric GFP-Fc and mCherry-Fc fusion proteins activate heterodimeric synthetic cytokine receptor complexes

Next, heterodimeric GFP-mCherry-Fc fusion proteins were generated resulting in a heterotetrameric GFP-mCherry-Fc-Fc-mCherry-GFP orientation (Fig 3A). GFP-mCherry-Fc was expressed in CHO-K1 cells and purified via Protein A-sepharose chromatography (Fig 3B). The overall yield of the GFP-mCherry-Fc fusion protein was 1.36 mg from 1 l cell culture supernatant. Non-reducing SDS-PAGE of the purified protein revealed the presence of dimeric, as well as higher order species (S2A Fig). Following purification, the main band observed in Western blotting analysis (Fig 3B) was in line with the expected molecular weight for intact GFP-mCherry-Fc. However, some degradation products became visible by Western blotting, supporting our previous finding, that GFP-mCherry fusion proteins were less stable.

First of all, Ba/F3-gp130, Ba/F3-gp130-(GFP$_{VHH}$-gp130), Ba/F3-gp130-(mCherry$_{VHH}$-gp130) and Ba/F3-gp130-(GFP$_{VHH}$-gp130)-(mCherry$_{VHH}$-gp130) cells (Fig 3C) were stimulated with purified GFP-Fc, mCherry-Fc or GFP-mCherry-Fc or Hyper-IL-6 demonstrating that all Fc-fusion proteins were selective with respect to synthetic cytokine receptor activation (Fig 3D). Importantly, this experiment suggested that GFP-mCherry-Fc was not able to efficiently activate Ba/F3-gp130-(GFP$_{VHH}$-gp130) cells compared to GFP-Fc, which was likely caused by the greater distance of GFP in GFP-mCherry-Fc compared to GFP-Fc. The greater distance of GFP in GFP-mCherry-Fc compared to GFP-Fc might simply not be sufficient to result in close juxtaposition of the synthetic receptors needed for effective receptor activation. This was supported by the determination of the half-maximal effective concentration to induce proliferation of Ba/F3-gp130-(GFP$_{VHH}$-gp130), Ba/F3-gp130-(mCherry$_{VHH}$-gp130) and Ba/

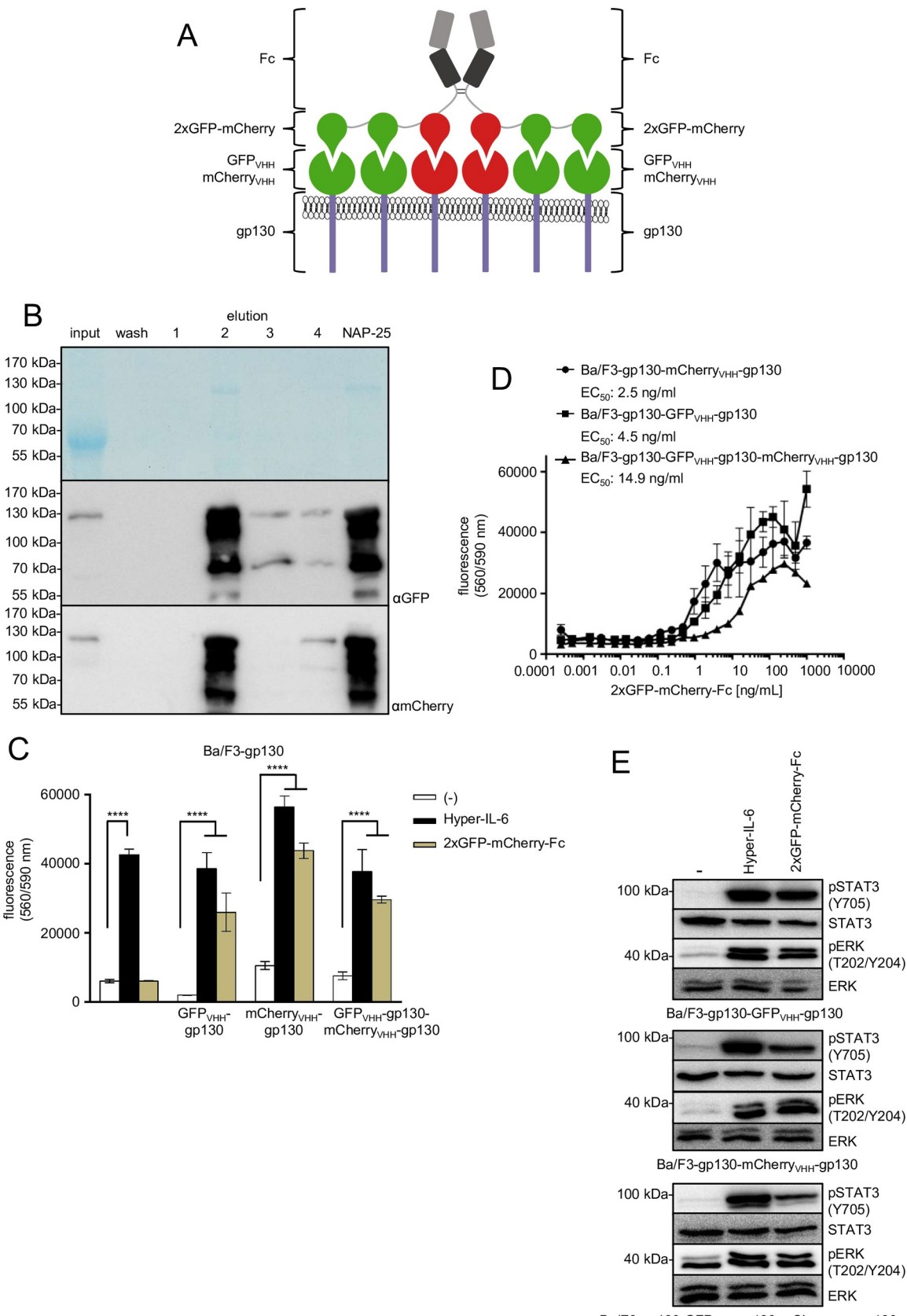

**Fig 4. Purification of a synthetic GFP-GFP-mCherry-Fc cytokine and analysis of signal transduction induced through GFP$_{VHH}$-gp130 and mCherry$_{VHH}$-gp130 SyCyRs.** (A) Schematic illustration of a GFP-GFP-mCherry fusion protein covalently linked to the Fc domain of an IgG antibody. The Fc domain forms a dimer and this leads to a hexameric GFP-GFP-mCherry assembly with two mCherry proteins in the center and two GFP subunits on the outside. The resulting protein binds GFP$_{VHH}$-gp130 and mCherry$_{VHH}$-gp130 SyCyRs or both. The image is similar to the original image and for illustrative purposes only. (B) Coomassie gel and Western blot analysis with GFP and mCherry antibodies. CHO-K1 cells were stably transduced with the cDNA coding for GFP-GFP-mCherry-Fc fusion protein. The supernatant containing the secreted protein with an expected molecular weight of 111.9 kDa was collected and purified via Protein A affinity chromatography. (C) Proliferation of Ba/F3-gp130, Ba/F3-gp130-(GFP$_{VHH}$-gp130), Ba/F3-gp130-(mCherry$_{VHH}$-gp130) and Ba/F3-gp130-(GFP$_{VHH}$-gp130)-(mCherry$_{VHH}$-gp130) cells was analyzed without cytokine (-), in presence of 10 ng/ml Hyper-IL-6 or in presence of 100 ng/ml 2xGFP-mCherry-Fc. One representative experiment out of four is shown. (D) Proliferation of Ba/F3-gp130-(GFP$_{VHH}$-gp130), Ba/F3-gp130-(mCherry$_{VHH}$-gp130) and Ba/F3-gp130-(GFP$_{VHH}$-gp130)-(mCherry$_{VHH}$-gp130) cells with increasing concentrations of 0.0004–1000 ng/ml 2xGFP-mCherry-Fc. One representative experiment out of three is shown. (E) STAT3 and ERK1/2 activation in Ba/F3-gp130-(GFP$_{VHH}$-gp130), Ba/F3-gp130-(mCherry$_{VHH}$-gp130) and Ba/F3-gp130-(GFP$_{VHH}$-gp130)-(mCherry$_{VHH}$-gp130) cells treated either with 10 ng/ml Hyper-IL-6 or 100 ng/ml 2xGFP-mCherry-Fc for 15 min. Equal amounts of protein (50 µg per lane) were analyzed via specific antibodies detecting phospho-STAT3/ERK1/2 and STAT3/ERK1/2. Western blot data shows one representative experiment out of four.

F3-gp130-(GFP$_{VHH}$-gp130)-(mCherry$_{VHH}$-gp130) cells which were 43.9 ng/ml, 1.97 ng/ml and 2.05 ng/ml, respectively (Fig 3E). This showed that GFP-mCherry-Fc was about 22fold less effective to induce proliferation of Ba/F3-gp130-(GFP$_{VHH}$-gp130) cells compared to Ba/F3-gp130-(mCherry$_{VHH}$-gp130) and Ba/F3-gp130-(GFP$_{VHH}$-gp130)-(mCherry$_{VHH}$-gp130) cells. Next, STAT3 and ERK phosphorylation of Ba/F3-gp130-(GFP$_{VHH}$-gp130), Ba/F3-gp130-(mCherry$_{VHH}$-gp130) and Ba/F3-gp130-(GFP$_{VHH}$-gp130)-(mCherry$_{VHH}$-gp130) cells induced by GFP-Fc, mCherry-Fc, GFP-mCherry-Fc or Hyper-IL-6 was compared (Fig 3F). Basically, all synthetic cytokines were able to activate STAT3 and ERK phosphorylation in a highly comparable manner, again supporting our finding that GFP-mCherry-Fc fusion proteins have high activity and selectivity. Interestingly, STAT3 and ERK activation of Ba/F3-gp130-(GFP$_{VHH}$-gp130) by GFP-mCherry-Fc was minor in comparison to GFP-Fc. This finding further supported our hypothesis that close juxtaposition of the synthetic receptors is not efficiently induced by GFP-mCherry-Fc.

Moreover, heterodimeric GFP-GFP-mCherry-Fc fusion proteins were generated resulting in heterohexameric assemblies composed of two central mCherry and apiece two GFP subunits on the outside (Fig 4A). GFP-GFP-mCherry-Fc was also expressed in CHO-K1 cells and purified via Protein A-sepharose chromatography (Fig 4B). An overall yield of 1.74 mg of the dimeric GFP-GFP-mCherry-Fc protein was obtained from 1 l cell culture supernatant (S2A Fig). Following purification, Coomassie staining revealed a main protein species in the preparation corresponding to GFP-GFP-mCherry-Fc. In addition, some truncated fusion proteins were detected by Western blotting. To examine the biological activity this synthetic ligand, Ba/F3-gp130, Ba/F3-gp130-(gp130-GFP$_{VHH}$), Ba/F3-gp130-(mCherry$_{VHH}$-gp130) and Ba/F3-gp130-(GFP$_{VHH}$-gp130)-(mCherry$_{VHH}$-gp130) cells were stimulated with purified GFP-GFP-mCherry-Fc or Hyper-IL-6 (Fig 4C). GFP-GFP-mCherry-Fc induced proliferation of cells expressing any synthetic cytokine receptor (Fig 4C). However, proliferation of these cell lines was induced with different EC$_{50s}$ (Fig 4D), 2.5 ng/ml for Ba/F3-gp130-(mCherry$_{VHH}$-gp130), 4.5 ng/ml for Ba/F3-gp130-(GFP$_{VHH}$-gp130) and 14.9 ng/ml for Ba/F3-gp130-(GFP$_{VHH}$-gp130)-(mCherry$_{VHH}$-gp130) cells. STAT3 and ERK phosphorylation of Ba/F3-gp130-(GFP$_{VHH}$-GFP), Ba/F3-gp130-(mCherry$_{VHH}$-gp130) and Ba/F3-gp130-(GFP$_{VHH}$-gp130)-(mCherry$_{VHH}$-gp130) cells induced by GFP-GFP-mCherry-Fc was comparable to the effect of Hyper-IL-6 (Fig 4E).

Our results suggest that the positioning of GFP and mCherry subunits in the multimeric assembly is critical for efficient receptor activation. Whereas hexameric GFP-GFP-mCherry-Fc-Fc-mCherry-GFP-GFP fusion proteins were able to efficiently activate GFP$_{VHH}$-gp130 and mCherry$_{VHH}$-gp130 synthetic cytokine receptors, this was not the case for tetrameric GFP-mCherry-Fc-Fc-mCherry-GFP fusion proteins which were only able to efficiently activate

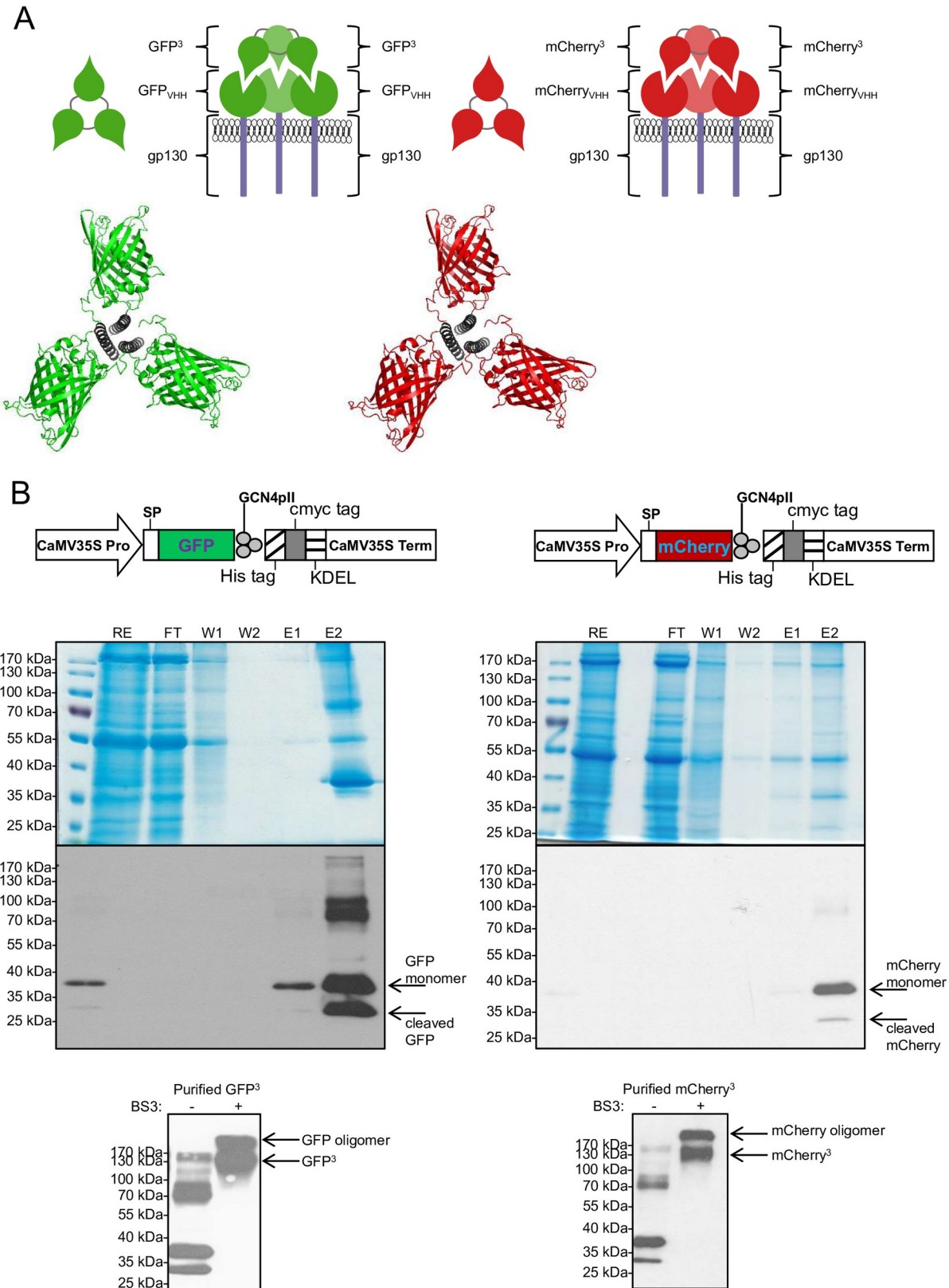

**Fig 5. Purification of synthetic GFP and mCherry (GFP³ and mCherry³) trimers expressed in *N. benthamiana* plants.** (A) Schematic illustration and modeling of GFP and mCherry trimers and the binding of GFP³ and mCherry³ to three GFP_VHH-gp130 or mCherry_VHH-

gp130 SyCyRs. Modeling images were created with UCFS Chimera. The image is similar to the original image and for illustrative purposes only. (B) Expression, purification and characterization of plant-derived GFP³ and mCherry³. GFP³ and mCherry³ fusion proteins were trimerized through c-terminal fusion of trimerization motif (GCN4pII) (12). Recombinant proteins also contained a c-myc tag to allow for downstream detection by Western blot, and a His-tag to facilitate their purification by IMAC. The legumine B4 signal peptide and the KDEL motif were used to promote transgene products retention in the endoplasmic reticulum. CaMV35S Pro: Cauliflower mosaic virus 35S ubiquituous promoter; CaMV 35S Term: Cauliflower mosaic virus 35S terminator. GFP proteins were purified from plant extract, analyzed in Coomassie gel and Western blot with myc antibody. Oligomeric state of trimeric eGFP was analyzed by crosslinking reaction, followed by SDS-PAGE and Western blotting. RE: plant raw extract; FT: flow-through; W1 and W2: wash fraction; E1 and E2: elution fraction; BS3: crosslinker.

mCherry$_{VHH}$-gp130 but not GFP$_{VHH}$-gp130 synthetic cytokine receptors. This is likely based on the fact that cytokine receptors need to be in close juxtaposition for efficient receptor activation [18], which is not the case for the two distant GFPs in GFP-mCherry-Fc. The design of the fusion protein might therefore be able to fine tune the activation of two synthetic cytokine receptor systems by stimulation with GFP-mCherry-Fc, one receptor will be fully activated via mCherry-dimers while the other is only minimally activated by distant GFP-dimers.

## Homotrimeric assembly of monomeric GFP and mCherry by GCN4pII result in highly effective synthetic cytokine receptor ligands

Finally, we used a trimeric GCN4pII motif to generate homotrimeric GFP and mCherry complexes. Again, we used Ba/F3 cells, expressing GFP$_{VHH}$-gp130, mCherry$_{VHH}$-gp130 and/or gp130 to analyze the biological activity of trimeric GFP (GFP³) and mCherry (mCherry³) (Fig 5A and 5B, S1C Fig). GFP³ and mCherry³ were transiently expressed in *Nicotiana benthamiana* leaves, and purified by immobilized metal affinity chromatography (IMAC), with an overall yield of 200 and 5 mg/kg leaf for GFP³ and mCherry³ (Fig 5B). In denaturing SDS-PAGE analysis, purified GFP³ and mCherry³ was predominantly found as monomers and a minority as cleaved monomers (Fig 5B). To determine the oligomeric state of purified plant-derived GFP³ and mCherry³ proteins, a crosslinking reaction was performed using a BS3 crosslinker. BS3 is a water-soluble and homo-bifunctional cross-linker which reacts with primary amines of target proteins to form stable amide bonds. When GFP³ and mCherry³ are exposed to BS3, amide bond-crosslinks between each subunit of a multimeric protein are formed. Crosslinked proteins were analyzed via SDS-PAGE under reducing and denaturing conditions, followed by Western blotting and subsequent immunodetection using anti-c-myc monoclonal antibody. As shown in Fig 5B, two bands with apparent molecular weights of between 130–170 kDa and over 170 kDa corresponding to trimers and oligomers (Fig 5B, +BS3 lanes) were observed. This result implies, that the native structure of plant-derived GFP³ and mCherry³ are mixtures of trimers and higher order oligomers.

Again, proliferation of Ba/F3-gp130 was only induced by Hyper-IL-6 but not by GFP³ or mCherry³, whereas GFP³ induced proliferation of Ba/F3-gp130-(GFP$_{VHH}$-gp130) cells and mCherry³ of Ba/F3-gp130-(mCherry$_{VHH}$-gp130) cells (Fig 6A). The EC$_{50}$ values of 0.58 ng/ml for GFP³ (Fig 6B), and 0.37 ng/ml for mCherry³ were determined on Ba/F3-gp130-(GFP$_{VHH}$-gp130) and Ba/F3-gp130-(mCherry$_{VHH}$-gp130) cells, respectively. STAT3 and ERK1/2 phosphorylation was induced in comparable degree by Hyper-IL-6, GFP³ and mCherry³ (Fig 6C).

In conclusion, we demonstrated the GFP³ and mCherry³ proteins were the most effective activators of our synthetic cytokine system.

## Conclusion

Recently, we developed a synthetic cytokine receptor system using synthetic ligands which facilitate high selectively and specificity and phenocopy endogenous cytokine signaling. The

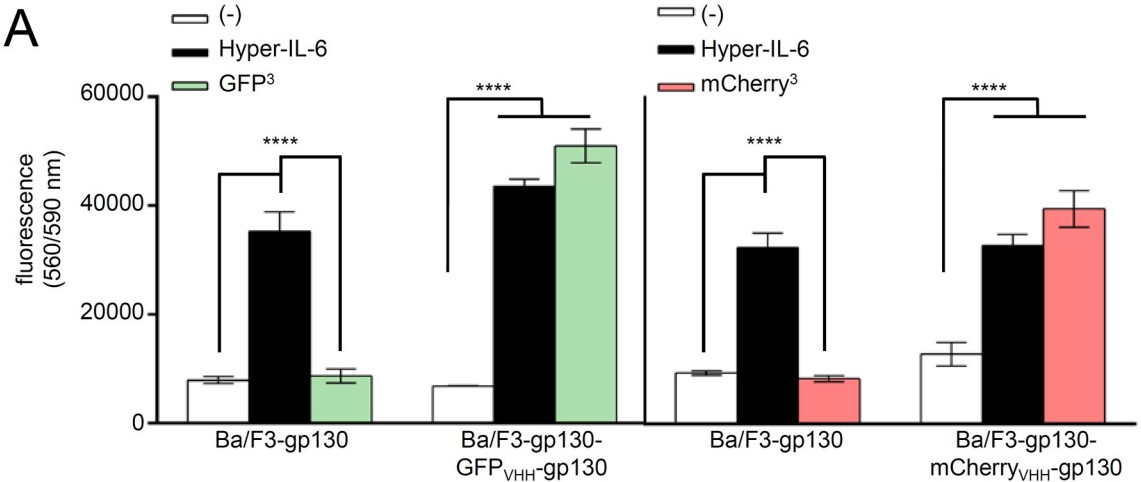

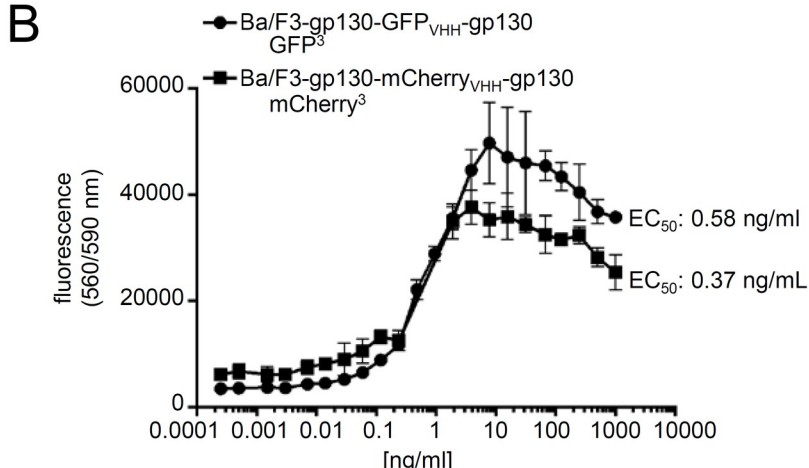

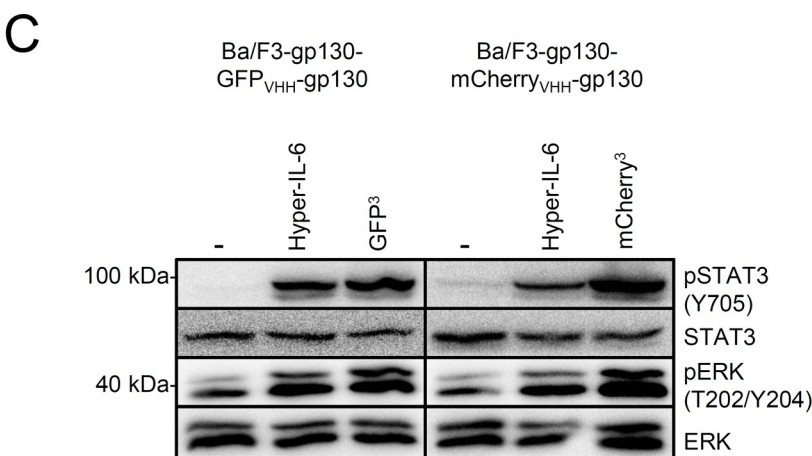

**Fig 6. Biological activity of GFP³ and mCherry³ on gp130-SyCyR cells.** (A) Proliferation of Ba/F3-gp130, Ba/F3-gp130-(GFP$_{VHH}$-gp130) and Ba/F3-gp130-(mCherry$_{VHH}$-gp130) cells without cytokine (-), in presence of 10 ng/ml Hyper-IL-6 or in presence of 100 ng/ml GFP³ or mCherry³. One representative experiment out of three is shown. (B) Proliferation of Ba/F3-gp130-(GFP$_{VHH}$-gp130) and Ba/F3-gp130-(mCherry$_{VHH}$-gp130) cells with increasing concentrations of 0.0004–1000 ng/ml GFP³ or mCherry³. One representative experiment out of three is shown. (C) STAT3 and ERK1/2 activation in Ba/F3-gp130-(GFP$_{VHH}$-gp130) and Ba/F3-gp130-(mCherry$_{VHH}$-gp130) cells treated either with 10 ng/ml Hyper-IL-6 or 100 ng/ml GFP³ or mCherry³ for 15 minutes. Equal

amounts of protein (50 µg per lane) were analyzed via specific antibodies detecting phosphor-STAT3/ERK1/2 and STAT3/ERK1/2. Western blot data shows one representative experiment out of three.

tailor-made activation of cytokine signaling has the potential to become a crucial add-on in CAR T-cell therapy [7]. Hence background-free, selective activation of cytokine receptors, as shown by our system, might become more and more important. Here, we showed that utilization of Fc domains of human IgG for dimerization of the synthetic monomeric ligands enabled the resulting fusion proteins to induce synthetic signal transduction. Further, our results imply that the distance of the subunits complex formed through Fc-mediated dimerization is crucial for effective receptor activation. Further our fully synthetic ligand-receptor system will enable the analysis of new hetero-/homo-cytokine receptor pairs or even higher ordered receptor clusters. Therefore, this could be an addition to existing synthekines which force non-natural receptor combinations in a juxtaposition and thereby achieve non-natural unique signaling pathways [6].

## Supporting information

**S1 Fig.** (A) Amino acid sequences of synthetic GFP$_{VHH}$- and mCherry$_{VHH}$-gp130 SyCyRs. From N- to C-terminus GFP$_{VHH}$-gp130: signal peptide of IL-11R (blue), myc-tag (orange), GFP$_{VHH}$ (green) and 13 aa of the ECD, TM and ICD gp130 (bold). From N- to C-terminus mCherry$_{VHH}$-gp130: signal peptide of IL-11R (blue), FLAGG-tag (green), HA-tag (purple), mCherry$_{VHH}$ (red) and 13 aa of the ECD, TM and ICD gp130 (bold). (B) Amino acid sequences of synthetic cytokines GFP-Fc, mCherry-Fc, GFP-mCherry-Fc and GFP-GFP-mCherry-Fc fusion proteins. From N- to C-terminus: signal peptide of IL-11R (blue), FLAGG-tag (red), GFP (green) or mCherry (purple) or both, TEV protease cleavage side (bold) and the Fc domain (orange). (C). Amino acid sequences of synthetic cytokines mCherry$^3$ and GFP$^3$. From N- to C-terminus: mCherry (red), GFP (green) trimeric GCNpII motif (brown), 6xHis tag (purple), myc tag (blue).
(TIFF)

**S2 Fig.** (A) Coomassie gel (left) and Western blot (right) analysis with anti-Fc and anti-mCherry of all purified proteins under reducing and non-reducing conditions. For Coomassie staining 5 µg, for western blot 5 ng of protein were loaded. (B) Proliferation of Ba/F3-gp130 cells with increasing concentrations of 0.0004–1000 ng/ml purified Hyper-IL-6-Fc or Hyper-IL-6 from CHO-K1 cell culture supernatants. One representative experiment out of three is shown.
(TIFF)

**S3 Fig.** Coomassie staining of the purification procedure of (A) 2xmCherry, (B) 2xGFP, (C) 3xmCherry, (D) 3xGFP, (E) GFP-mCherry and (F) GFP-GFP-mCherry proteins expressed in *E. coli*. The image is similar to the original image and for illustrative purposes only.
(TIF)

**S4 Fig.** (A) CHO-K1 cells were stably transduced with the respective cDNA coding for 2xGFP, 3xGFP. The supernatants were collected and expression analyzed via western blot using anti-GFP antibodies. The respective supernatants were either analyzed directly or after storage at 4˚C for 5 d. The image is similar to the original image and for illustrative purposes only. (B) CHO-K1 cells were stably transduced with the respective cDNA coding for GFP-mCherry, GFP-GFP-mCherry, 2xmCherry and 3xmCherry. The supernatants were collected and the expression analyzed via western blot using anti-mCherry antibodies. The respective

supernatants were either analyzed directly or after storage at 4˚C for 5 d. The image is similar to the original image and for illustrative purposes only.
(TIF)

## Acknowledgments

We thank Petra Oprée for her assistance.

## Author Contributions

**Conceptualization:** Sofie Mossner, Jens M. Moll, Udo Conrad, Jürgen Scheller.

**Funding acquisition:** Jürgen Scheller.

**Methodology:** Sofie Mossner, Hoang T. Phan, Saskia Triller.

**Project administration:** Jürgen Scheller.

**Visualization:** Jens M. Moll.

**Writing – original draft:** Sofie Mossner, Udo Conrad, Jürgen Scheller.

**Writing – review & editing:** Jürgen Scheller.

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
