## [Decision Letter · Decision Letter 0]

25 Feb 2020

PONE-D-19-35000

Multimerization strategies for efficient production and purification of highly active synthetic cytokine receptor ligands

PLOS ONE

Dear Dr. Scheller,

Thank you for submitting your manuscript to PLOS ONE. We found only a very minor issues in the manuscvript and, therefore, invite you to submit a revised version of the manuscript that addresses the points raised the reviewer.

We would appreciate receiving your revised manuscript by Apr 10 2020 11:59PM. To enhance the reproducibility of your results, we recommend that if applicable you deposit your laboratory protocols in protocols.io, where a protocol can be assigned its own identifier (DOI) such that it can be cited independently in the future. For instructions see: http://journals.plos.org/plosone/s/submission-guidelines#loc-laboratory-protocols

We look forward to receiving your revised manuscript.

Kind regards,

Marek Cebecauer

Academic Editor

PLOS ONE

Journal Requirements:

Reviewers' comments:

Reviewer's Responses to Questions

**Comments to the Author**

1. Is the manuscript technically sound, and do the data support the conclusions?

Reviewer #1: Yes

2. Has the statistical analysis been performed appropriately and rigorously? 

Reviewer #1: Yes

3. Have the authors made all data underlying the findings in their manuscript fully available?

Reviewer #1: Yes

4. Is the manuscript presented in an intelligible fashion and written in standard English?

Reviewer #1: Yes

5. Review Comments to the Author

Reviewer #1: I think the authors have made the point that different synthetic heterodimeric or heterotrimeric can be used for receptor activation.

I suggest the authors to add a "conclusion" to the manuscript to allow the reader to have a short take home message and a look into the future.

Some spelling mistakes are in the ms, maybe a native English speaker should go through the text before publication.

6. PLOS authors have the option to publish the peer review history of their article (what does this mean?). If published, this will include your full peer review and any attached files.

Reviewer #1: No

---

## [Author Response · Author response to Decision Letter 0]

4 Mar 2020

Reviewer 1: 

I think the authors have made the point that different synthetic heterodimeric or heterotrimeric can be used for receptor activation.

I suggest the authors to add a "conclusion" to the manuscript to allow the reader to have a short take home message and a look into the future.

Some spelling mistakes are in the ms, maybe a native English speaker should go through the text before publication.

Author´s answer: 

We have carefully revised the manuscript and have corrected spelling mistakes. 

Additionally, we have added a “Conclusion”-section at the end of the manuscript.

---

## [Editor Report · Decision Letter 1]

10 Mar 2020

Multimerization strategies for efficient production and purification of highly active synthetic cytokine receptor ligands

PONE-D-19-35000R1

Dear Dr. Scheller,

We are pleased to inform you that your manuscript has been judged scientifically suitable for publication and will be formally accepted for publication once it complies with all outstanding technical requirements.

With kind regards,

Marek Cebecauer

Academic Editor

PLOS ONE
---

## [Editor Report · Acceptance letter]

12 Mar 2020

PONE-D-19-35000R1 

Multimerization strategies for efficient production and purification of highly active synthetic cytokine receptor ligands 

Dear Dr. Scheller:

I am pleased to inform you that your manuscript has been deemed suitable for publication in PLOS ONE. Congratulations! Your manuscript is now with our production department. 

With kind regards,

on behalf of

Mr Marek Cebecauer 

Academic Editor

PLOS ONE